# Bond Strength of Universal Adhesives to Dentin: A Systematic Review and Meta-Analysis

**DOI:** 10.3390/polym13050814

**Published:** 2021-03-07

**Authors:** Louis Hardan, Rim Bourgi, Naji Kharouf, Davide Mancino, Maciej Zarow, Natalia Jakubowicz, Youssef Haikel, Carlos Enrique Cuevas-Suárez

**Affiliations:** 1Department of Restorative Dentistry, School of Dentistry, Saint-Joseph University, 1107 2180 Beirut, Lebanon; louis.hardan@usj.edu.lb (L.H.); rim.bourgi@net.usj.edu.lb (R.B.); 2Department of Biomaterials and Bioengineering, INSERM UMR_S 1121, Biomaterials and Bioengineering, 67000 Strasbourg, France; endodontiefrancaise@outlook.com (D.M.); youssef.haikel@unistra.fr (Y.H.); 3Department of Endodontics, Faculty of Dental Medicine, Strasbourg University, 67000 Strasbourg, France; 4Private Practice, “NZOZ SPS Dentist” Dental Clinic and Postgraduate Course Centre, pl. Inwalidow 7/5, 30-033 Cracow, Poland; dentist@dentist.com.pl (M.Z.); nljakubowicz@gmail.com (N.J.); 5Dental Materials Laboratory, Academic Area of Dentistry, Autonomous University of Hidalgo State, Circuito Ex Hacienda La Concepción S/N, San Agustín Tlaxiaca, Hidalgo 42160, Mexico

**Keywords:** adhesion, dental adhesive, dental bonding, dentin bonding agent, universal adhesive

## Abstract

Currently, the availability of a wide variety of universal adhesives makes it difficult for clinicians to choose the correct system for specific bonding situations to dentin substrate. This study aimed to determine whether there are any alternative techniques or additional strategies available to enhance the bond strength of universal adhesives to dentin through a systematic review and meta-analysis. Two reviewers executed a literature search up to September 2020 in four electronic databases: PubMed, ISI Web of Science, Scopus, and EMBASE. Only in vitro studies that reported the dentin bond strength of universal adhesives using additional strategies were included. An analysis was carried out using Review Manager Software version 5.3.5 (The Nordic Cochrane Centre, The Cochrane Collaboration, Copenhagen, Denmark). The methodological quality of each in vitro study was assessed according to the parameters of a previous systematic. A total of 5671 potentially relevant studies were identified. After title and abstract examination, 74 studies remained in systematic review. From these, a total of 61 studies were included in the meta-analysis. The bond strength of universal adhesives to dentin was improved by the use of one of the following techniques: Previous application of matrix metalloproteinases (MMP) inhibitors (*p* < 0.001), prolonged application time (*p* = 0.007), scrubbing technique (*p* < 0.001), selective dentin etching (*p* < 0.001), non-atmospheric plasma (*p* = 0.01), ethanol-wet bonding (*p* < 0.01), prolonged blowing time (*p* = 0.02), multiple layer application (*p* = 0.005), prolonged curing time (*p* = 0.006), and hydrophobic layer coating (*p* < 0.001). On the other hand, the use of a shortened application time (*p* = 0.006), and dentin desensitizers (*p* = 0.01) impaired the bond strength of universal adhesives to dentin. Most of the analyses performed showed a high heterogenicity. The in vitro evidence suggests that the application of universal adhesives using some alternative techniques or additional strategies may be beneficial for improving their bonding performance to dentin. This research received no external funding. Considering that this systematic review was carried out only with in vitro studies, registration was not performed.

## 1. Introduction

Despite the enormous improvements made in adhesive technology throughout the last 50 years, there are still some unresolved problems regarding the durability of the adhesive interface [1]. Impervious seal between resin and dentin substrate remains difficult to achieve [2], and it is doubtful that an ideal interdiffusion of the adhesive system inside collagen scaffold can be produced in a harmonious fashion [3]. In fact, hydrolytic degradation combined by the action of dentinal matrix enzymes are thought to destabilize the tooth-adhesive bond and break down the unprotected collagen fibrils [4]. Even though sufficient resin–dentin bonding is generally immediately reached, lessened bonding efficiency arises with time [5].

All in all, the actual mechanisms of degradation are far from being fully understood. In this sense, it is necessary to strictly characterize the recommendations of each company and to follow them; otherwise, deterioration of the adhesive could occur [6]. Achieving adequate hybridization within collagen fibrils and the stability of resin-dentin interface are of key importance [7]. It is fundamental to recognize that the mechanism of hybridization in which an interdiffusion zone, also called: “Hybrid layer”, is created, fulfills the occurrence of the micromechanical retention of the restoration [8].

The quality of adhesion relies on numerous factors and may vary according to the adhesive system used which can be either the self-etch or the etch-and-rinse [9]. A simplification of the classical concept of the dental bonding by means of a less-sensitive technique, faster application, and various optional applicability is possible nowadays with the introduction of universal adhesives [10]. Based on this, universal adhesives constitute one type of all-in-one adhesives with the presence of both hydrophobic and hydrophilic components on the same bottle [11], and according to manufacturers, they can be used in both direct and indirect substrates [12]. However, the stability and durability of dentin-adhesive interface generated by these new multimode adhesive systems continue to be questionable [13]. One of the major concerns of these systems was related to the increase of nanoleakage after aging, and thus leading to a limited bond durability [14]. The issue is that until now, there has been no golden pattern protocol for achieving stable and optimal adhesion of universal adhesive systems to dentin.

Furthermore, improvement in bond strength can be realized by means of numerous strategies, and therefore, suggested by many authors [15,16]. Thus, the aim of this study was to evaluate whether, in vitro, the immediate and long-term bonding performance of universal adhesives to dentin would be improved by the use of any additional strategies through a systematic review and meta-analysis. The null hypothesis of the present study was that there would be no difference in immediate and long-term bond strength to dentin substrate when using universal adhesives with any alternative technique other than the instructions of the manufacturer.

## 2. Materials and Methods

This systematic review and meta-analysis was performed in accordance with the PRISMA 2020 statement [17]. The following PICOS framework was used: Population, permanent dentin substrate (human); intervention, application of universal adhesives with additional steps or different techniques; control, application of universal adhesives according to manufacturer instructions; outcome, bond strength; and study design, in vitro studies. The research question was: “Is there any technique or additional step available to improve the bond strength of universal adhesives to dentin”?

### 2.1. Literature Search

The literature search was conducted by two independent reviewers (C.E.C.-S. and R.B.) up to 26 September 2020. The following four electronic databases were screened to identify manuscripts that could be included: PubMed (MedLine), ISI Web of Science, Scopus, and Embase. The inter-examiner agreement was quantified using the kappa coefficient. The keywords and search strategy used in PubMed are listed in Table 1. The full search strategy for ISI Web of Science, Scopus, and EMBASE databases is presented as Appendix A. The reviewers also performed a hand search of reference lists of included articles for identification of additional papers. Following the initial screening, all studies were imported into Mendeley Desktop 1.17.11 software (Glyph & Cog, LLC, London, UK) to eliminate duplicates.

### 2.2. Study Selection

Two reviewers (R.B. and C.E.C.-S.) individually assessed the titles and abstracts of all studies. Manuscripts for full-text reviews were selected according to the following eligibility criteria: (1) In vitro studies reporting the effect of the use of additional steps for the application of universal adhesive systems in the bond strength to dentin; (2) evaluating the bond strength of universal adhesive systems to dentin with a resin-based material as an antagonist; (3) including a control group in which universal adhesive was applied according to manufacturers’ instructions; (4) including mean and standard deviation (SD) data in MPa on shear, micro-shear, tensile, and micro-tensile bond tests. Only manuscripts published in the English language were considered. Manuscripts that involved different substrates further than those recognized in the inclusion criteria were not considered. Case reports, pilot studies, case series, and reviews were also excluded. Full copies of all of the potentially relevant studies were assessed. Papers that seemed to meet the inclusion criteria or had insufficient data in the title and abstract to produce a clear decision were designated for full analysis. The full-text manuscripts were considered independently in duplicate by two review authors. Any discrepancy concerning the eligibility of the included studies was decided and resolved through discussion and agreement by a third reviewer (L.H.). Only papers that satisfied the eligibility criteria listed were included.

### 2.3. Data Extraction

Data of concern from the involved studies were extracted using Microsoft Office Excel 2019 (Microsoft Corporation, Redmond, WA, USA). These data comprised the year of publication, type of tooth storage, strategy evaluated for bonding to dentin, type of bond strength test evaluated, and adhesive system used. If any information was missing, the corresponding authors of the included studies were notified twice via an e-mail to retrieve the missing data. If the authors did not respond within 2 weeks of the first contact, the missing information was not included.

### 2.4. Quality Assessment

The methodological quality of each in vitro study was assessed by two reviewers (L.H. and R.B.), according to the parameters of a previous systematic review [16]. The risk of bias in each article was evaluated according to the description of the following parameters: Specimen randomization; single-operator protocol implementation; blinding of the testing machine operator; the presence of a control group; standardization of the sample preparation; failure mode evaluation; use of all materials according to the manufacturer’s instructions; and description of the sample size calculation. If the authors reported the parameter, the study received a “YES” for that specific parameter. In the case of missing information, the parameter received a “NO.” The risk of bias was classified according to the sum of “YES” answers received: 1 to 3 indicated a high bias, 4 to 6 medium, and 7 to 8 indicated a low risk of bias.

### 2.5. Statistical Analysis

Meta-analysis was performed using Review Manager Software version 5.3.5 (The Nordic Cochrane Centre, The Cochrane Collaboration, Copenhagen, Denmark). The analysis was carried out using the random-effects model and pooled-effect estimates were obtained by comparing the standardized mean difference between bond strength values obtained from the control and experimental. In addition, immediate and long-term bond strength data were analyzed separately. In studies where several experimental groups were compared with the same control group, data from the experimental groups (mean, standard deviation, and sample size) were combined for the meta-analysis [18]. A *p*-value < 0.05 was considered statistically significant. Statistical heterogeneity of the treatment effect among studies was assessed using the Cochran Q test and the inconsistency I^2^ test.

## 3. Results

A total of 5671 publications were retrieved in all databases. A flowchart that summarizes the study selection process according to the PRISMA Statement is shown in Figure 1. The literature review retrieved 4303 manuscripts for the initial examination after the duplicates were removed. Of these, 4198 studies were excluded after reviewing the titles and abstracts. In total, 105 studies were examined by full-text reading. Of these, 31 studies were excluded [19,20,21,22,23,24,25,26,27,28,29,30,31,32,33,34,35,36,37,38,39,40,41,42,43,44,45,46,47], the reasons for exclusion are mentioned in the Appendix A, totalizing 74 articles in the qualitative analysis. Of these, 13 studies were excluded: In 9 studies, there were not enough study groups to perform any comparison [39,40,48,49,50,51,52,53,54], 2 studies did not have any control group [38,55], 1 study combined data from different generations of adhesives [56], and 1 study did not present the mean and SD [57]. Thus, 61 studies were included in the meta-analysis. The inter-examiner agreement was excellent (kappa coefficient = 0.87).

The qualitative synthesis of the studies included in this systematic review are summarized in Table 2. Several strategies were identified: Air abrasion [58,59], application of crosslinking agents [57,60,61,62,63,64,65], application of dentin desensitizers [66,67], use of matrix metalloproteinases (MMP) inhibitors [10,68,69,70,71,72,73,74,75,76,77], the use of the ethanol-wet bonding technique [78], application of an extra hydrophobic resin layer [56,79,80,81,82,83,84,85], dentin pretreatment with laser [55,86,87,88,89,90,91,92], multiple layer application [93,94,95,96,97,98,99], non-atmospheric plasma [100,101], prolonged curing time [102,103], prolonged blowing time [104,105,106], application of the adhesive using a scrubbing technique [107,108,109], selective dentin etching [110,111,112], prolonged application time [104,105,113], shortened application time [99,106,114,115,116], application of the adhesive using an electric-current [48], combining of phosphoric acid with chlorhexidine [39], using ozone as cavity disinfectant [50], pre-warming of adhesive bottles [52], air-blowing with warm-air [53] smear layer deproteinization [54], and removal of the oxygen inhibited layer [40].

Separate analysis for each technique was performed. The main results of the datasets evaluated are shown in Figure 2, Figure 3, Figure 4, Figure 5, Figure 6, Figure 7, Figure 8, Figure 9, Figure 10, Figure 11, Figure 12, Figure 13, Figure 14, Figure 15 and Figure 16. The bond strength of universal adhesives to dentin was improved when the following techniques were used: Previous application of MMP inhibitors (*p* < 0.001), prolonged application time (*p* = 0.007), scrubbing technique (*p* < 0.001), selective dentin etching (*p* < 0.001), non-atmospheric plasma (*p* = 0.01), ethanol-wet bonding (*p* < 0.01), prolonged blowing time (*p* = 0.02), multiple layer application (*p* < 0.001), prolonged curing time (*p* = 0.006), and hydrophobic layer coating (*p* < 0.001).

On the other hand, the use of a shortened application time (*p* < 0.001) and dentin desensitizers (*p* = 0.01) impaired the bond strength of universal adhesives to dentin. Finally, the use of lasers (*p* = 0.06), air abrasion (*p* = 0.016), and crosslinking agents (*p* = 0.08) prior to the application of a universal adhesive did not have any effect on the bond strength.

According to the parameters for methodological quality assessment, most studies included were classified with medium risk of bias (Figure 17). Most of the studies analyzed failed to report the single operator, operator blinded, and sample size calculation parameters.

## 4. Discussion

A systematic review and meta-analysis was conducted regarding the bond strength of universal adhesives to dentin depending on the additional strategy used: Extra hydrophobic coat, prolonged curing time, multiple layer application, prolonged blowing time, dentin desensitizer, air abrasion, ethanol-wet bonding technique, non-thermal atmospheric plasma, selective dentin etching for 3 s, scrubbing technique, laser technology, shortened application time, prolonged application time, MMP inhibitors, and crosslinking agents. Some of the techniques mentioned above improved the bond strength of the universal adhesives to dentin, while others lacked any effect, and many others impaired the bond strength. Therefore, the hypothesis tested in this study was partially accepted.

For an extra hydrophobic coat application, both immediate and aged bond strength were improved. One should bear in mind that the bond durability of universal adhesives to dentin might be limited by both their thinnest film thickness and thin hybrid layer [117]. Nevertheless, the clinical benefit of a thicker adhesive layer is still scant in literature. In this regard, the placement of an extra hydrophobic coat has been undoubtedly used recently to improve the bonding performance of universal adhesives to dentin [118,119], by means of a higher hydrophobicity, and superior polymerization [79,80]. Moreover, better sealing of the adhesive interface, which would indicate better bond degradation prevention, could be possible by the additional of a hydrophobic layer [120]. The aforementioned statement could be explained by the phenomena in which the fluid flow across the adhesive interface decreases due to the increase in the adhesive layer thickness [81,121]. Numerous studies advocated the fact that the short and long-term bonding performance of one-step-self-etch adhesive systems could be improved by placement of such an extra layer [120,122]. In this respect, aging stability could be material dependent [79], and the use of an additional hydrophobic coat could be more beneficial for self-etch rather than the etch-and-rinse strategy [81]. Overall, the meta-analysis showed that the extra layer of hydrophobic coat supplements unsolvated monomers with hydrophobic nature into adhesive interface, which subsequently reduces the concentration of unreacted monomers and retained solvents in the adhesive layer [123], increasing the bond strength of the adhesive interface [124].

Immediate and aged bond strength were increased by increasing the photoactivation time. Ideally, enough energy must be applied by a restorative material to convert monomers into stable polymers [103]. To improve polymerization and reduce permeability, increasing the time of light exposure of adhesives beyond the manufacturer’s recommendation has been suggested [86]. This technique has been proved to guarantee an adequate degree of conversion, [110] making the adhesive less prone to water sorption [125]. In addition, extended curing time could maintain the camphorquinone in a state which allows it to react with a co-initiator, producing a higher energy density and increasing free radical formation [126]. In this manner, the exposure time recommended by the respective manufacturer for adhesive systems is not sufficient to obtain an optimal polymerization [125,127]. Hence, a longer exposure time up to 40 s is preferred, as was demonstrated by this meta-analysis.

Immediate bond strength is increased by using multiple layer application; however, this effect could be not observed after aging. Previous studies suggested that doubling the number of adhesive layers improves bond strength by enhancing monomer penetration into hybrid layers and increasing chemical interactions [93,128]. Therefore, an additional layer application should be considered as a crucial clinical step. Almost all universal adhesives were based on the functional monomer 10-MDP (methacryloyloxydecyl dihydrogen phosphate) imperative to obtain stable nanolayer structure (10-MDP/Ca salts) in the hybrid layer and adhesive layer [11]. MDP monomer needs an appropriate time of 20 s for its chemical interaction to take place; however, applying a second coat of such a monomer without curing the first one permits the first layer to sufficiently interact with hydroxyapatite and thus promotes supplementary bonding [129].

Immediate bond strength was improved by prolonged blowing time; however, this effect could be not observed after aging. In the bonding procedure, air-drying for solvent evaporation is considered a crucial step after the application of the adhesive. Since ethanol and water can form hydrogen bonds with each other and also with the monomer, it is difficult to remove water from ethanol-based adhesives compared to those of acetone-based adhesives [130]. In a study by Saikaew et al., longer air-blowing time caused significant increases in the bond strength of ethanol-based adhesives, whereas the bond strength of acetone-based adhesive was not affected by the different air-blowing time. This conclusion seems to partially support the results in this meta-analysis, as improvement in bond strength was not observed in all adhesives after aging. In this sense, prolonged air-blowing time for 15–30 s could be suggested in order to enhance immediate adhesive properties of some universal adhesives [113].

The use of dentin desensitizer impaired both immediate and aged bond strength. The application of dentin desensitizers aids in occluding dentinal tubules and its combination with adhesive systems could alleviate the post-operative sensitivity accompanying the placement of composite resin [131]. However, its influence on bonding performance by means of diverse bonding agents remains to be assessed as the desensitized treated dentin might be harmful for bonding [132]. This could be explained by the formation of nanoparticle layers distributed inside dentin substrates which impede the interaction of adhesive to demineralized dentin [133]. Additionally, resin tags formation might be disturbed by tubule occlusion with calcium-containing desensitizer pastes, resulting in diminished bond strength [134].

The use of air abrasion did not have any significant effect on both immediate and aged bond strength. Despite this, it’s important to note that by modifying dentin with airborne particles may negatively affect the chemical interaction between functional monomers and hydroxyapatite. This could be explained by the remnant particles resulting from the air abrasion application [29]. Dentin might experience damage by harder particles like alumina particles as a result of its lower elastic modulus and lower hardness [135]. Furthermore, these particles are able to chemically and/or mechanically alter the collagen fibers, which hinders the adhesive penetration and consequently lessens the bond strength [32].

Immediate bond strength was improved by the use of ethanol-wet bonding technique. Ethanol is acknowledged to be a solvent of choice when compared to water since it is able to reduce the diameter of collagen fibrils and increase the interfibrillar space, allowing monomers to easily infiltrate the collagen fibrils [136]. This technique, called “ethanol-wet bonding technique”, has been proven to efficiently seal the dentinal matrix, reducing the dentin-resin interface permeability, which would reduce the activity of collagenolytic enzymes and consequently improve bond durability to dentin substrate [137]. In this simplified technique, 100% of ethanol is applied for 1 min, which could be clinically acceptable [138], when collagens are successfully protected with ethanol, greater longevity of adhesive interface is observed, creating a stable bond [138].

The use of non-thermal atmospheric plasma improves the aged bond strength, but presents no effect on the immediate bond strength. Plasma has been reported to improve bonding to dentin [139]. It is described as partially ionized gases that comprise electronically excited atoms, molecules, ions, and free radical species [100]. These particles are considered as reactive crosslinkers, able to form chemical functional groups on the surface of dentin [139]. The use of non-thermal atmospheric plasma proved to increase dentin wettability [140,141], improve resin polymerization [142], and enhance resin infiltration [143,144]. Another potential explanation for the bond-promotion influence is that non-thermal atmospheric plasma stimulates the dentin substrate by leaving free radicals or peroxides, thus strengthening the interaction between collagen fibrils and resin monomers [145]. Furthermore, a reduction in the expression and activity of MMP-2 and MMP-9 could be possible by applying a non-atmospheric plasma on thyroid papillary cancer cells [103].

Both immediate and aged bond strength were improved by the application of selective dentin etching for 3 s. Van Meerbeek et al. suggested that protecting collagen from hydrolysis and early degradation of the bond could be achieved by keeping hydroxyapatite around collagen fibrils [146]. Therefore, a functional monomer such as (10-MDP), which is found inside universal adhesives, can form stable calcium-phosphate complexes and self-assemble into the form of a regular layered structure at the apatite surface. Maintaining calcium at the bonded interface may favor this uncharacteristic chemical bonding process. Consequently, in order to improve the resin-dentin bonding, a new approach called “Selective dentin etching” has emerged. This technique is based on the application of phosphoric acid during 3 s in dentin, which after rinsing and drying, leaves a partially demineralized substrate [110,111,112]. In general, the remaining calcium-content ratio was dependent on the time of etching; the following residual calcium-content ratios were found to be similar on self-etch mode and 3 s of selectively etched dentin. Etching times above 5 s considerably lessen these ratios [111]. This finding clearly indicates that this technique has been proven as an alternative to improve the long-term bond strength of universal adhesives to dentin.

The use of scrubbing technique improves both immediate and aged bond strength. Active application of universal adhesives enhanced dentin bond durability of the etching mode by facilitating the penetration of adhesives into the branches of dentin tubules, and improving solvent evaporation [109]. Actually, the active application of adhesives using a scrubbing technique leads to the impregnation of a higher rate of monomers inside the smear layer, hence improving adhesive-interface quality and decreasing hybrid layer degradation [147,148]. After a manual force is applied through an adhesive rubbing motion into the dentin substrate, dentin acts similar to a sponge, and then the dentin collagen network is compressed. Compressed collagen enlarges as the pressure is relieved and the infiltration of the adhesive monomers into collagen network is enhanced [149]. Furthermore, since monomers like 10-MDP with hydrophobic nature are well impregnated due to adhesive agitation, aging couldn’t negatively affect bonding performance [37,150].

Immediate bond strength was impaired by the use of lasers, while aged bond strength showed no statistical difference. Overall, there was no difference between control and laser. Laser technology was initiated in the 20th century, to be an alternative technique to the conventional drilling technique [151]. Lasers had the ability to absorb water and hydroxyapatite necessary for hard tissue removal [152]. Higher absorption during laser ablation could be the reason for carious removal by means of micro-explosion, water vaporization, and heating [153]. A clear substrate, free of debris, could be achieved by such a laser, but considering adhesion to dentin, numerous studies revealed a decrease in bond strength with the use of Er:YAG laser [154,155]. This could be explained by the fact that when applying a laser, water and organic constituents of dentin vaporize, producing the thermomechanical ablation of the inorganic constituents [156]. Collagen networks fuse by dentinal ablation, causing a lack of interfibrillar space, and accordingly, resin monomers will not be able to penetrate adequately, affecting the bond strength negatively [157].

The choice between shortened application time and prolonged application time may direct dentists toward the shortened application time. However, this study’s results showed that the shortened application time impairs both immediate and aged bond strength. Manufacturers of universal adhesives simplify the application technique, nevertheless, Saikaew et al. (2016) revealed certain weaknesses with reduced application time on immediate dentin bond strength [116]. This could be in agreement with the finding of this study and could be explained by the fact that three factors affected the reduced application time, these factors being: Smear layer interaction, method of application, and adhesive composition. These factors could be considered for each adhesive [106]. Acetone had a high vapor pressure compared to water and ethanol, so a longer application time for acetone-based adhesive could enhance the evaporation of such a solvent and increase bond strength; however, this is not the case with ethanol-based adhesives, since water inside universal adhesives forms a hydrogen bond with ethanol and with the shortened application time, the ethanol evaporation might be affected and the bond strength stability diminishes [130]. Accordingly, the residual water and solvent might compromise the polymerization of adhesive, which could accelerate monomer degradation with time and impair bond strength [158]. This statement is in agreement with this meta-analysis. With regards to smear layer interaction, shortened application time may not dissolve the smear layer easily, and the bond strength could be compromised [106,115]. The longer the adhesive-smear layer interaction, the better the penetration of monomers through the smear layer to form a stronger bond with the dentin substrate [106,159].

Prolonged application time improves immediate bond strength, while aged bond strength showed no statistical difference. With longer application time of adhesives, a better performance of resin-dentin bonds could be achieved, and monomers are able to diffuse inward [128]. This is responsible for increasing the saturation of collagen by resin since monomers ideally should fill the space between the exposed collagen fibrils [160,161], otherwise adhesive durability could be affected. In addition, when the adhesive is applied for a prolonged period of time, more solvent can evaporate, hence permitting the development of a stronger polymer within the dentin substrate and higher resin-dentin bond strength [96,162]. Therefore, increasing the application time enhances the chemical reaction between functional monomers and hydroxyapatite, which is considered a crucial step for increasing monomer infiltration [163].

MMP inhibitors improve immediate and long-term bond strength of universal adhesive to dentin. Regardless of the adhesive system used, a major problem is the degradation of the hybrid layer over time. It is well known that the dental adhesion process is based on the creation of a suitable and compact hybrid layer created by impregnating the dentin substrate with monomers, rather on the morphology or the thickness of the hybrid layer [164], in general, the more compact and homogeneous the hybrid layer, the better the stability of the resin–dentin bond [165]. When a portion of collagen remained unprotected within the hybrid layer, a cleavage by endogenous and exogenous collagenolytic enzymes of the family endopeptidases, like the matrix metalloproteinases (MMPs), can be assumed [7]. In order to improve the quality of the resin–dentin interface, some studies focused on counteracting enzymatic biodegradation by the use of matrix metalloproteinases (MMPs) inhibitors [166]. The use of MMP inhibitors on dentin substrate had been advocated to improve the longevity of adhesive and to reduce aging over time by means of altering the active site of MMPs [123]. These inhibitors include chlorhexidine, 1-ethyl-3-(3-dimethylaminopropyl) carbodiimide, green tea extract, galardin, tetracyclines and analogues, quaternary ammonium salts, and ethylenediaminetetraacetic acid [167,168,169,170]. This conclusion seems to support the finding of this study, as the use of MMP inhibitors lessens interfacial aging over time by hindering the activation of endogenous dentin enzymes, which are dependable for the degradation of collagen network.

Immediate bond strength was not improved by the use of crosslinking agents, while aged bond strength showed an improvement. Overall, there was no difference between control and crosslinking agents. Since higher mechanical properties and lower biodegradation rates of collagen are needed, the use of collagen crosslinking agents in adhesive procedures have gained interest [5]. Covalent cross-links created with external cross-linkers such as ribose, glutaraldehyde, carbodiimide hydrochloride, proanthocyanidins, chlorhexidine, riboflavin, chitosan, polyphenols, and genipin have been found to be very stable, by offering significant advantages in developing mechanically stable collagen scaffolds [171]. Ideally, the introduction of additional cross-links prevents collagen molecules inside dentin from sliding past each other under stress [172], which increases the mechanical strength of collagen fibers and reduces extensibility [47]. Collagen crosslinking was initiated as a substitute dentin pre-treatment to strengthen dentinal collagen fibrils, as well as enhance the bonding durability [173,174]. This matches with the finding of this manuscript, as the effect of crosslinkers appear with time to reinforce collagen by means of maintenance of an expanded position which is able to receive solvents and monomers [175,176].

The methodological quality assessment revealed that most studies included were classified with medium risk of bias, which indicate that the quality of the evidence of the outcome evaluated could be high. In relation to this, it should be highlighted that blinding and sample size calculation were not reported in most of the studies analyzed, and failure to describe these parameters could increase the likelihood of performance, attrition, and detection bias [177].

From this review, in vitro evidence was evaluated with regards to the alternative techniques used in the literature for the bonding efficacy of universal adhesives to dentin substrate. The results of this review should be considered with caution since, in clinical situations, a wet environment, masticatory stresses, and pH, lead to a rapid adhesive-dentin interface. Future research must be conducted, especially randomized controlled clinical trials, with the purpose of gaining a better understanding of the performance of universal adhesives in the clinical success of resin-based restorations to dentin substrate.

## 5. Conclusions

With the limitation of a long distance between laboratory studies and clinical randomized evaluations, the in vitro evidence suggests that the application of universal adhesives using some alternative techniques or additional strategies different than the manufacturers’ recommendations may be beneficial for improving the bonding performance to dentin. Due to its practicality, the active application of the adhesive, together with the evaporation of the solvent for periods of time longer than 10 s, could be recommended for improving the bond strength values to dentin. In addition, prolonged curing time of up to 40 s, selective dentin etching for 3 s, double adhesive application time, extra hydrophobic resin layer, previous application of MMP inhibitors, non-thermal atmospheric plasma, ethanol-wet bonding strategy, and multiple layer application (2 layers and more) are recommended. On the other hand, shortened application time, and application of desensitizer to the dentin should be avoided since the bond strength could be impaired.

## Figures and Tables

**Figure 1 polymers-13-00814-f001:**
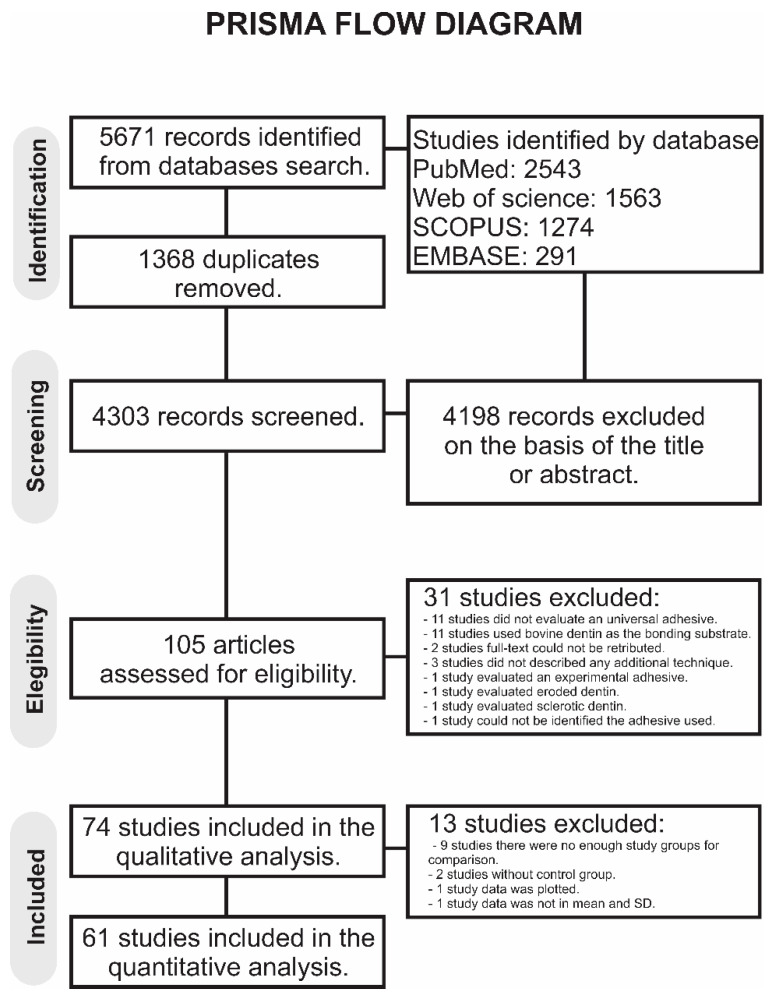
Prisma flow diagram of the study.

**Figure 2 polymers-13-00814-f002:**
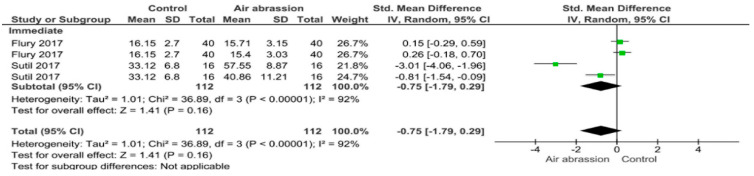
Results of the meta-analysis of immediate and aged bond strength of universal adhesive to dentin after air abrasion.

**Figure 3 polymers-13-00814-f003:**
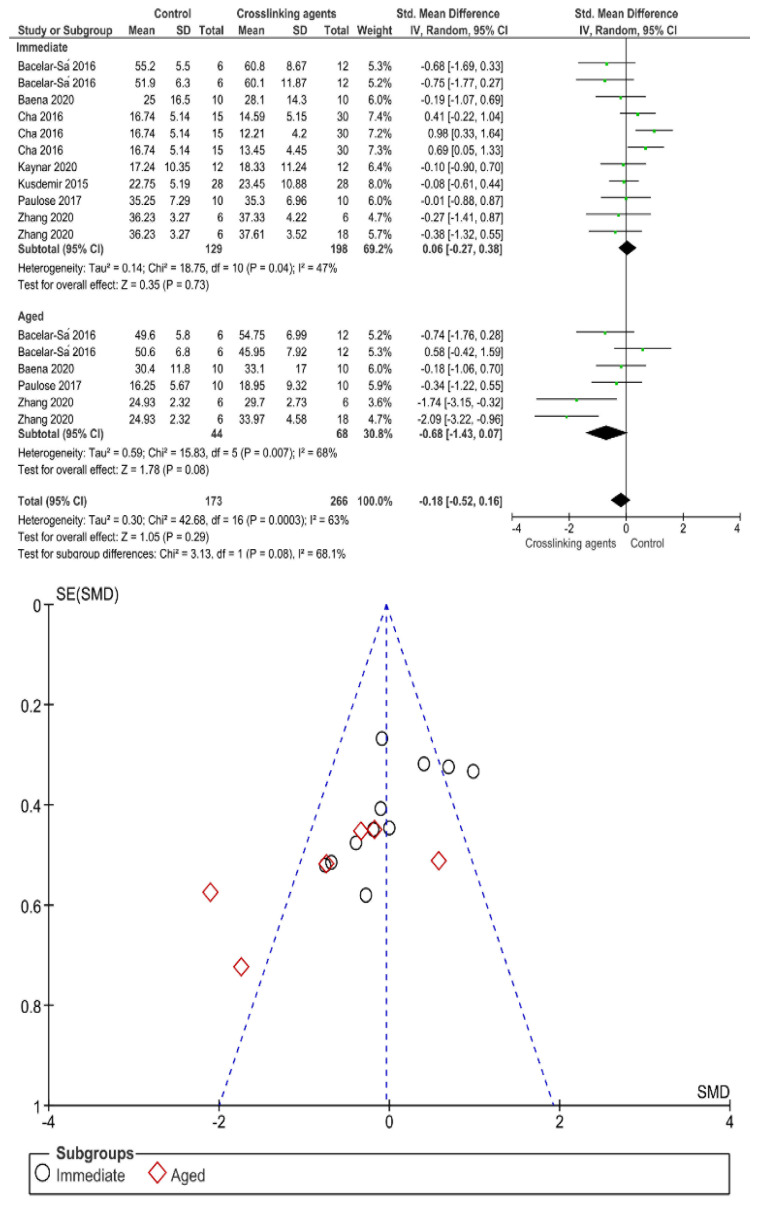
Results of the meta-analysis of immediate and aged bond strength of universal adhesive to dentin after application of crosslinking agents.

**Figure 4 polymers-13-00814-f004:**
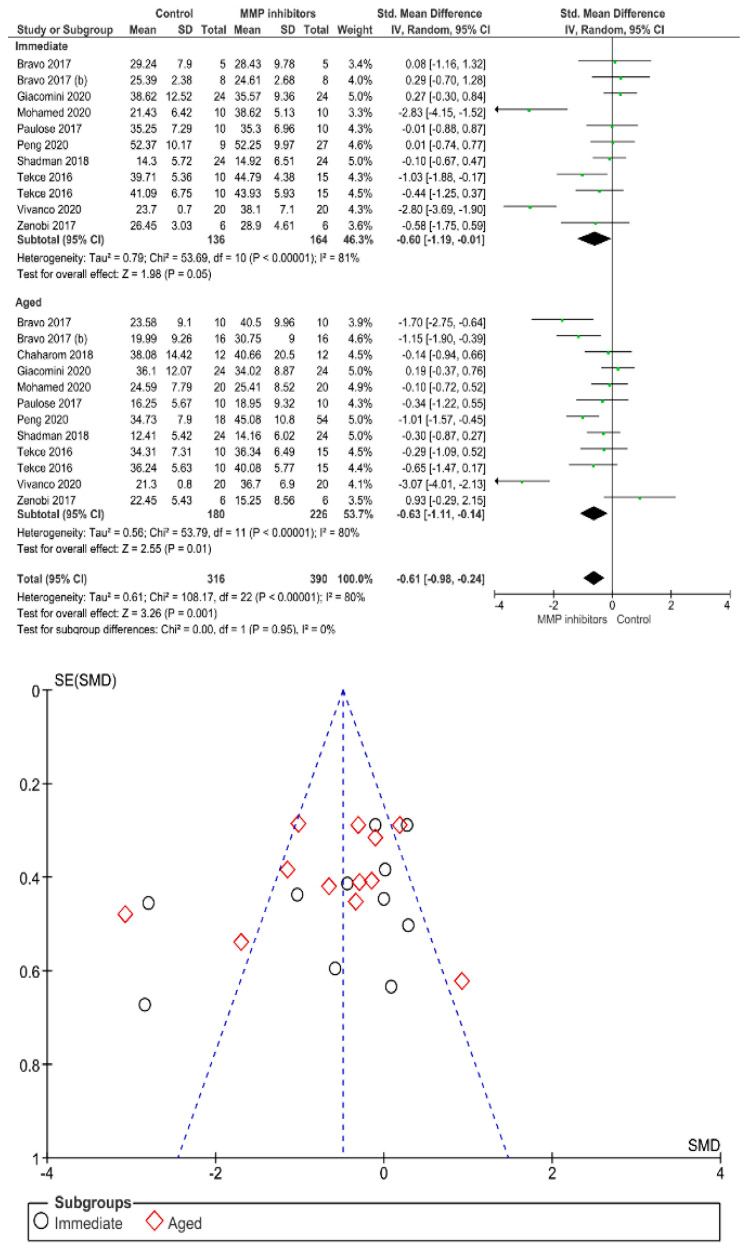
Results of the meta-analysis of immediate and aged bond strength of universal adhesive to dentin after application of MMP inhibitors.

**Figure 5 polymers-13-00814-f005:**
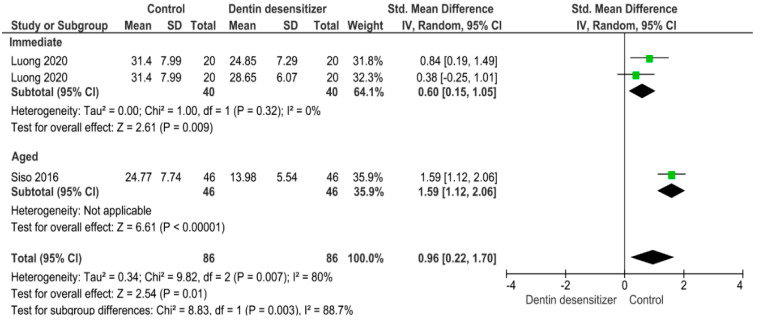
Results of the meta-analysis of immediate and aged bond strength of universal adhesive to dentin after application of dentin desensitizers.

**Figure 6 polymers-13-00814-f006:**
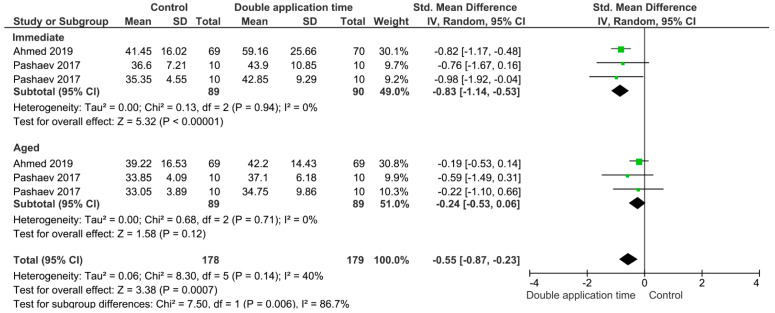
Results of the meta-analysis of immediate and aged bond strength of universal adhesive to dentin using the double application time technique.

**Figure 7 polymers-13-00814-f007:**
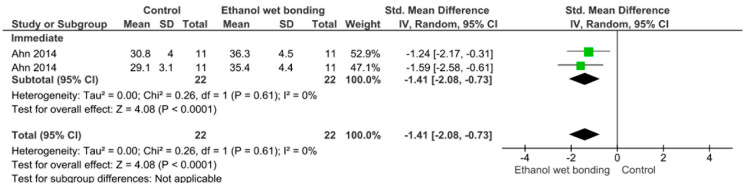
Results of the meta-analysis of immediate bond strength of universal adhesive to dentin using the ethanol-wet bonding technique.

**Figure 8 polymers-13-00814-f008:**
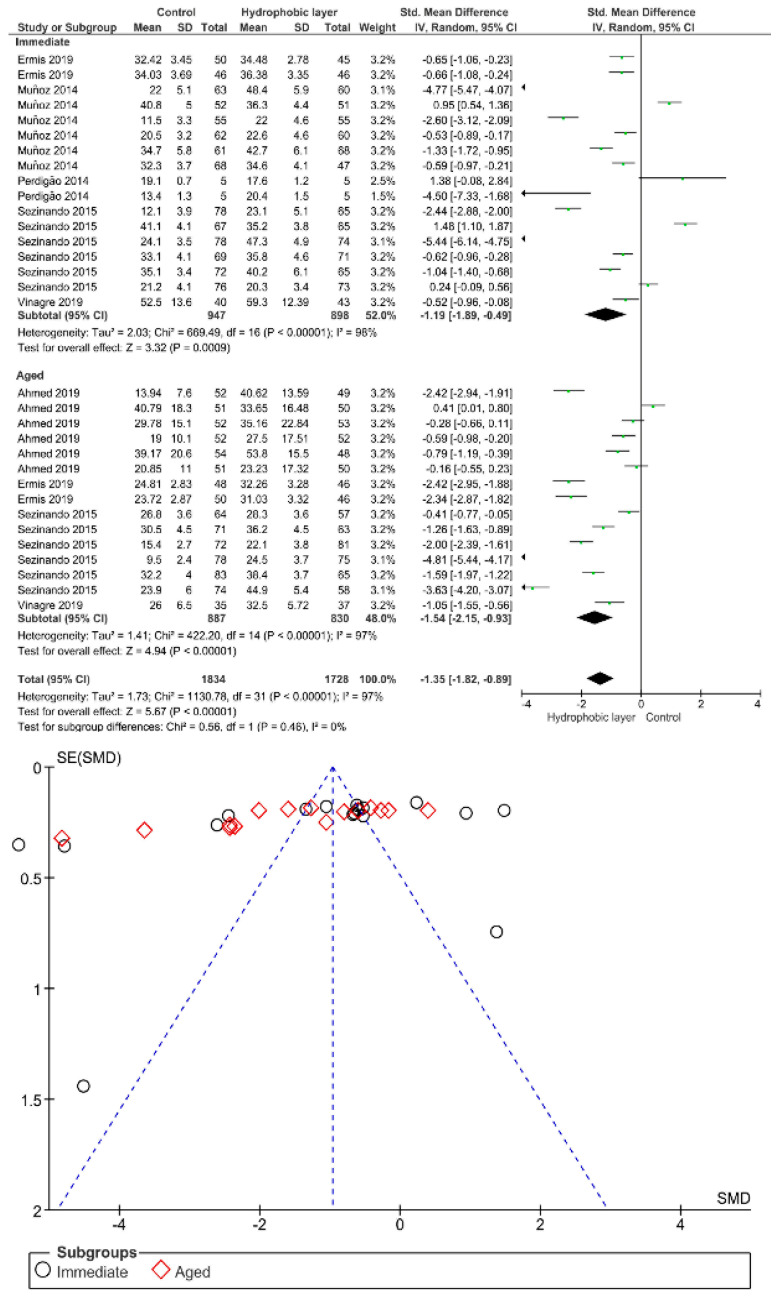
Results of the meta-analysis of immediate and aged bond strength of universal adhesive to dentin after the application of a hydrophobic layer.

**Figure 9 polymers-13-00814-f009:**
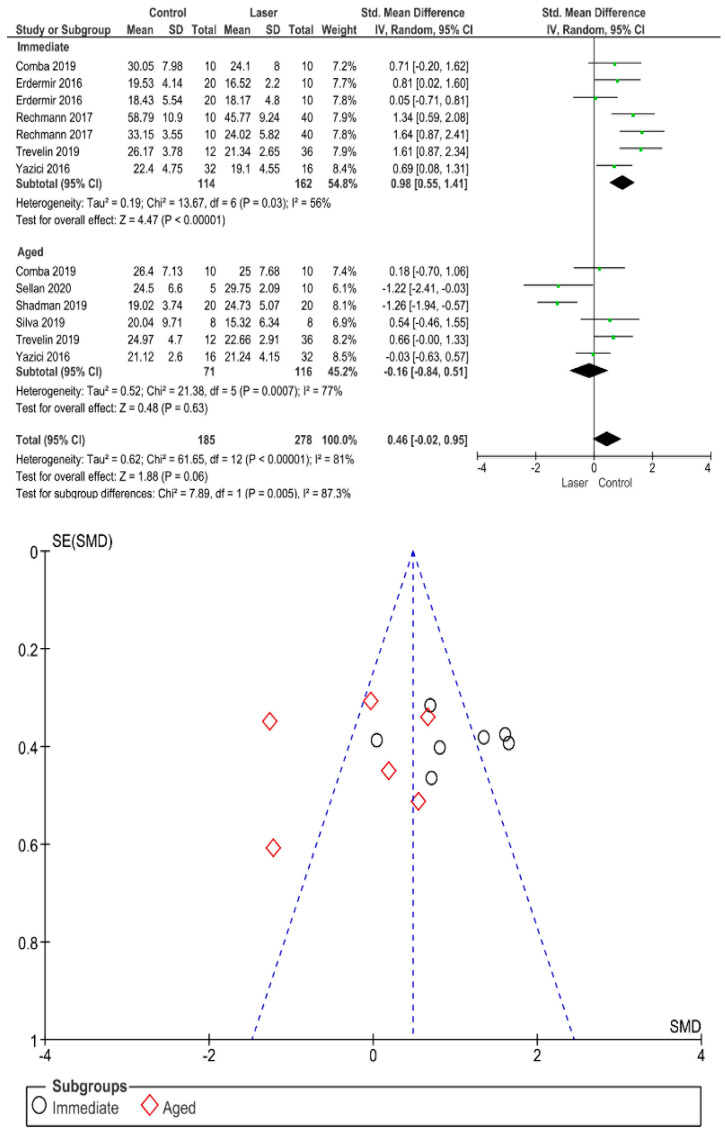
Results of the meta-analysis of immediate and aged bond strength of universal adhesive to dentin using a laser as surface pretreatment.

**Figure 10 polymers-13-00814-f010:**
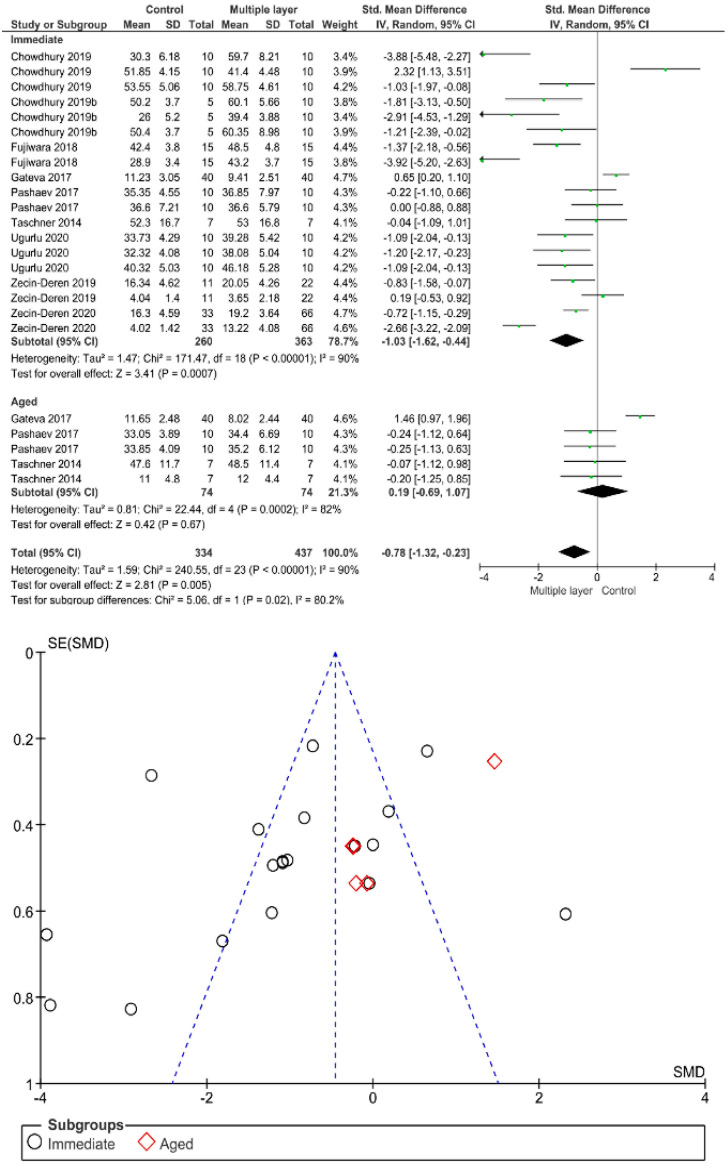
Results of the meta-analysis of immediate and aged bond strength of universal adhesive to dentin used in multiple layers.

**Figure 11 polymers-13-00814-f011:**
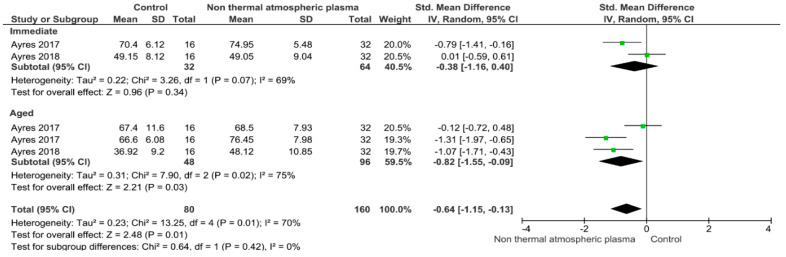
Results of the meta-analysis of immediate and aged bond strength of universal adhesive to dentin using non-atmospheric plasma.

**Figure 12 polymers-13-00814-f012:**
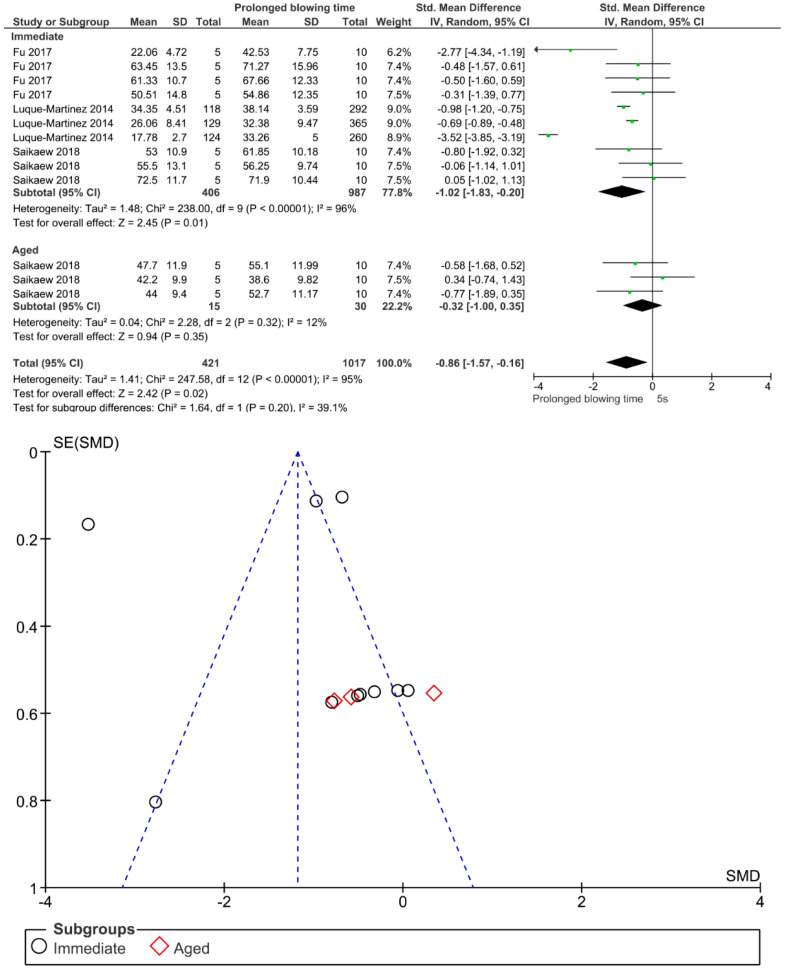
Results of the meta-analysis of immediate and aged bond strength of universal adhesive to dentin with prolonged solvent evaporation time.

**Figure 13 polymers-13-00814-f013:**
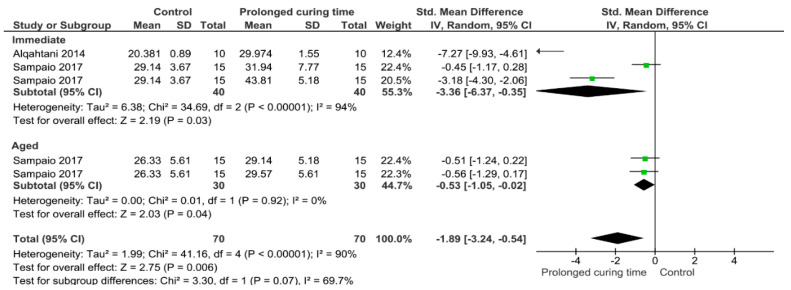
Results of the meta-analysis of immediate and aged bond strength of universal adhesive to dentin with prolonged curing time.

**Figure 14 polymers-13-00814-f014:**
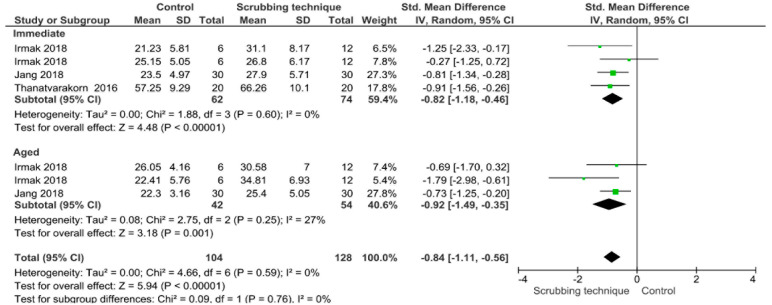
Results of the meta-analysis of immediate and aged bond strength of universal adhesive to dentin applied in an activated mode.

**Figure 15 polymers-13-00814-f015:**
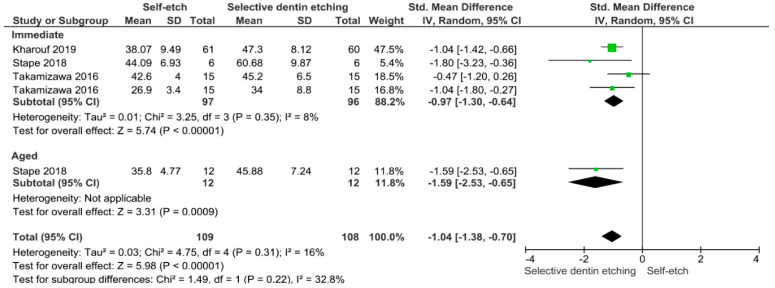
Results of the meta-analysis of immediate and aged bond strength of universal adhesive to dentin using the selective dentin etching.

**Figure 16 polymers-13-00814-f016:**
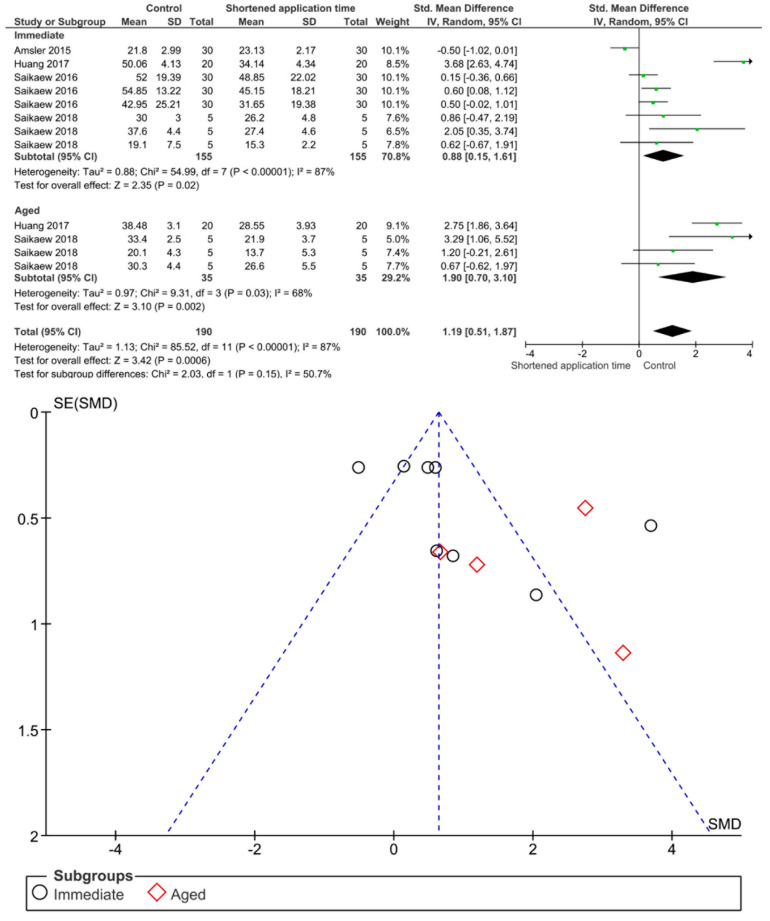
Results of the meta-analysis of immediate and aged bond strength of universal adhesive to dentin applied using a shortened application time.

**Figure 17 polymers-13-00814-f017:**
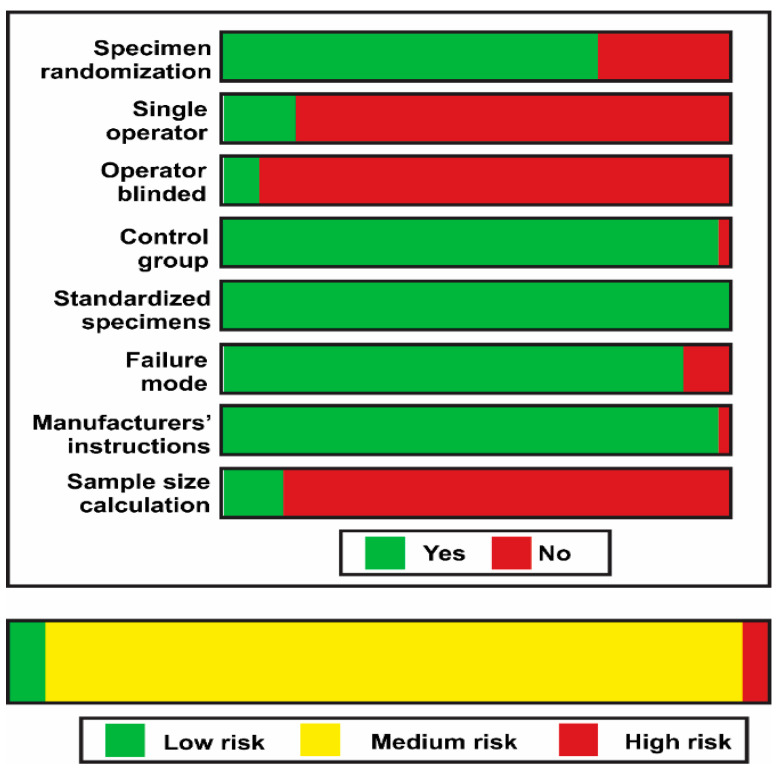
Summary of the results of the risk of bias assessment.

**Table 1 polymers-13-00814-t001:** Search strategy used in PubMed.

Search Strategy	
**# 1**	Universal adhesives OR Universal adhesive OR Universal simplified adhesive systems OR Universal Dental Adhesives OR Multipurpose adhesives OR multi-purpose adhesives OR multimode adhesives OR multi-mode adhesives or universal bonding agent
**# 2**	Bond OR Bonding OR Dental bonding OR Bonding efficacy OR bond strength OR Bonding performance OR bonding effectiveness OR Bond performance OR adhesive properties OR microtensile strength OR Micro-tensile strength OR bonding properties OR Microtensile bond strength OR shear bond strength OR microshear bond strength OR performance
**# 3**	Dentine OR Dentin
**# 4**	#1 and #2 and #3

**Table 2 polymers-13-00814-t002:** Demographic and study design data of the included studies.

Study	Aging/Storage	Strategy	Bond Strength Test	Adhesive System Used
Flury, 2017	24 h of water storage at 37 °C	Air Abrasion	Dentin μSBS	Scotchbond Universal (3M ESPE)
Sutil, 2017	24 h of water storage at 37 °C	Air Abrasion	Dentin μTBS	Scotchbond Universal (3M ESPE)
Bacelar-Sá, 2017	24 h of water storage at 37 °C 6 months of water storage at 37 °C	Application of collagen crosslinking agents	Dentin μTBS	Scotchbond Universal (3M ESPE) Prime & Bond Elect (Dentsply) All-Bond 3 (Bisco Inc.) G-Aenial (GC Corp.)
Baena, 2020	24 h of water storage at 37 °C 10,000 thermocycles between 5 and 55 °C	Application of collagen crosslinking agents	Dentin μTBS	OptiBond FL (Kerr) Scotchbond Universal (3M ESPE)
Cha, 2016	24 h of water storage at 37 °C	Application of collagen crosslinking agents	Dentin SBS	Scotchbond Universal (3M ESPE)
de Lima, 2018	24 h of water storage at 37 °C	Application of collagen crosslinking agents	Dentin μTBS	Prime & Bond Elect (Dentsply)
Kaynar, 2020	10,000 thermocycles between 5 and 55 °C	Application of collagen crosslinking agents	Dentin μTBS	Peak Universal Bond (Ultradent Products Inc.) G-Premio Bond (GC Corp)
Kusdemir, 2015	48 h of water storage at 37 °C	Application of collagen crosslinking agents	Dentin μTBS	Single Bond Universal (3M ESPE)
Paulose, 2017	24 h of water storage at 37 °C 1 year of water storage at 37 °C	Application of collagen crosslinking agents	Dentin μTBS	Single Bond Universal (3M ESPE) Adper Scotchbond Multi-Purpose Plus (3M ESPE)
Zhang, 2020	24 h of water storage at 37 °C 1-month collagenase ageing	Application of collagen crosslinking agents	Dentin μTBS	Single Bond Universal (3M ESPE)
Luong, 2020	1 h of water storage at 37 °C	Application of dentin desensitizer	Dentin μSBS	Parkell Universal Adhesive PBOND (Parkell, Edgewood, NY, USA) Clearfil SE 2 (Kuraray Noritake)
iso, 2016	1000 thermal cycles between 5 °C and 55 °C	Application of dentin desensitizer	Dentin μTBS	Clearfil Universal Bond (Kuraray Noritake)
Bravo, 2017	(72 h, 3 months, 6 months) of storage in distilled water at room temperature	Application of matrix metalloproteinases (MMP) inhibitors	Dentin μTBS	Adper Scotchbond 1XT (3M ESPE) Adper Prompt L-Pop (3M ESPE) Single Bond Universal (3M ESPE)
Bravo, 2017 (b)	(72 h, 3 months, 6 months) of storage in distilled water at room temperature	Application of MMP inhibitors	Dentin SBS	Adper Scotchbond 1XT (3M ESPE) Adper Prompt L-Pop (3M ESPE) Single Bond Universal (3M ESPE)
Chaharom, 2018	500 thermal cycles between 5 °C and 55 °C	Application of MMP inhibitors	Dentin μTBS	All-Bond 3 (Bisco Inc.) Clearfil SE Bond (Kuraray Noritake) All Bond Universal (Bisco Inc.)
Giacomini, 2020	24 h of storage in artificial saliva at 37 °C 6 months of storage in artificial saliva at 37 °C	Application of MMP inhibitors	Dentin μTBS	Single Bond Universal (3M ESPE) Adper Single Bond 2 (3M ESPE)
Mohamed, 2020	(24 h, 3 months, 6 months) of storage in distilled water at 37 °C	Application of MMP inhibitors	Dentin μTBS	Single Bond Universal (3M ESPE)
Paulose, 2017 (b)	24 h and 1 year of storage in distilled water at 37 °C	Application of MMP inhibitors	Dentin μTBS	Single Bond Universal (3M ESPE) Adper Single Bond 2 (3M ESPE)
Peng, 2020	24 h of water storage at 37 °C 10,000 cycles between 5 °C and 55 °C one-month collagenase ageing samples were immersed in 0.1 mg/mL collagenase solution at 37 °C	Application of MMP inhibitors	Dentin μTBS	Scotchbond Universal (SBU; 3M ESPE)
Shadman, 2018	24 h and 500 thermal cycles between 5 °C and 55 °C in distilled water at 37 °C	Application of MMP inhibitors	Dentin SBS	Scotchbond Universal (3M ESPE) Scotchbond Multi-Purpose Plus (3M ESPE)
Tekçe, 2016	24 h of water storage at 37 °C 12 months of water storage at 37 °C	Application of MMP inhibitors	Dentin μTBS	Single Bond Universal (3M ESPE) All Bond Universal (Bisco Inc.)
Vivanco, 2020	30 days of water storage at 37 °C 1.200.000 cycles with 5/37/55 °C	Application of MMP inhibitors	Dentin μTBS	Single Bond Universal (3M ESPE) Adper Scotchbond Multi-Purpose Plus (3M ESPE)
Zenobi, 2017	water immersion for 24 h 200,000 mechanical cycles under a load of 30 N, at a rate of 2 Hz for one week	Application of MMP inhibitors	Dentin μTBS	Single Bond Universal (3M ESPE)
Ahn, 2014	24 h of water storage at room temperature	Ethanol-wet bonding	Dentin μTBS	Clearfil SE Bond (Kuraray Noritake) G-aenial Bond (GC Corp) Xeno V (Dentsply) BeautiBond (Shofu Inc.) Adper Easy Bond (3M ESPE) Single Bond Universal (3M ESPE) All Bond Universal (Bisco Inc.)
Ahmed, 2019	1 month of water storage at 37 °C 25,000 and 50,000 thermocycles	Hydrophobic resin layer	Dentin μTBS	Clearfil SE Bond 2 (Kuraray Noritake, Osaka, Japan) Clearfil Universal Bond Quick (Kuraray Noritake) G-Premio Bond (GC Corp., Tokyo, Japan) Prime&Bond Active (Dentsply, Konstanz, Germany)
Chasqueira, 2020	24 h of water storage at 37 °C 6 months in a sodium azide solution (pH = 7, 37 °C) 18 months in a sodium azide solution (pH = 7, 37 °C)	Hydrophobic resin layer	Dentin SBS	Adper Scotchbond 1XT (3M ESPE) Clearfil S3 Bond Plus (Kuraray Noritake) Solobond M (Voco; Cuxhaven, Germany) Adper Easy Bond (3M ESPE) Scotchbond Universal (3M ESPE)
Ermis, 2019	24 h of water storage at 37 °C 6 months of water storage at 37 °C	Hydrophobic resin layer	Dentin μTBS	Clearfil SE Bond (Kuraray Noritake) OptiBond XTR (Kerr Co., Orange, CA, USA) Clearfil Universal Bond Quick (Kuraray Noritake) Single Bond Universal (3M ESPE, St Paul, MN, USA)
Muñoz, 2014	24 h of water storage at 37 °C	Hydrophobic resin layer	Dentin μTBS	All Bond Universal (Bisco Inc., Schaumburg, IL, USA) Scotchbond Universal (3M ESPE) G-Bond Plus (GC Corp.)
Perdigāo, 2014	24 h of water storage at 37 °C	Hydrophobic resin layer	Dentin μTBS Enamel μSBS	G-Bond Plus (GC Corp.)
Sezinando, 2015	24 h of water storage at 37 °C 6 months of water storage at 37 °C	Hydrophobic resin layer	Dentin μTBS	Scotchbond Universal (3M ESPE) G-Bond Plus (GC Corp.) All Bond Universal (Bisco Inc.)
Vinagre, 2019	7 days of water storage at 37 °C 4 years of water storage at 37 °C	Hydrophobic resin layer	Dentin μTBS	Scotchbond Universal (3M ESPE) Adper Scotchbond Multi-Purpose Plus (3M ESPE)
Comba, 2019	24 h of water storage at 37 °C 6 months of water storage at 37 °C	Laser	Dentin μTBS	Clearfil SE Bond 2 (Kuraray Noritake) All Bond Universal (Bisco Inc.) OptiBond FL (Kerr Co.)
Sellan, 2020	10,000 cycles between 5 °C and 55 °C	Laser	Dentin μTBS	Single Bond Universal (3M ESPE)
Shadman, 2019	500 cycles of thermocycling between 5 and 55 °C	Laser	Dentin SBS	Scotchbond Universal (3M ESPE)
Silva, 2016	48 h of water storage at 37 °C	Laser	Dentin μTBS	Scotchbond Universal (3M ESPE)
Silva, 2019	500 cycles of thermocycling between 5 and 55 °C	Laser	Dentin SBS	Clearfil SE Bond (Kuraray Noritake) Scotchbond Universal (3M ESPE)
Rechmann, 2017	24 h of water storage at 37 °C	Laser	Dentin SBS Enamel SBS	OptiBond Solo Plus (Kerr Co.) Peak Universal Bond (Ultradent Products Inc., South Jordan, UT, USA) Scotchbond Universal (3M ESPE)
Trevelin, 2019	24 h of water storage at 37 °C 1 year of storage in artificial saliva at 37 °C	Laser	Dentin μSBS	Scotchbond Universal (3M ESPE)
Yazici, 2016	24 h of water storage at 37 °C	Laser	Dentin SBS	Single Bond Universal (3M ESPE)
Chowdhury, 2019	24 h of water storage at 37 °C	Multiple layer application	Dentin μTBS	Scotchbond Universal (3M ESPE) G-Premio Bond (GC Corp) Clearfil Megabond 2 (Kuraray Noritake)
Chowdhury, 2019 (b)	24 h of water storage at 37 °C	Multiple layer application	Dentin μTBS	Scotchbond Universal (3M ESPE) G-Premio Bond (GC Corp) Clearfil Megabond 2 (Kuraray Noritake)
Fujiwara, 2018	24 h of water storage at 37 °C	Multiple layer application	Dentin and Enamel SBS	Scotchbond Universal (3M ESPE) Prime & Bond elect (Dentsply, Caulk Milford, DE, USA) G-aenial Bond (GC Corp.) Beautibond (Shofu Inc., Kyoto, Japan) OptiBond XTR (Kerr Co.)
Pashaev, 2017	24 h of water storage at 37 °C 6 months of water storage at 37 °C	Multiple layer application	Dentin μTBS	Single Bond Universal (3M ESPE) All Bond Universal (Bisco Inc.) Adper Easy One (3M ESPE) Adper Single Bond 2 (3M ESPE)
Taschner, 2014	24 h in artificial saliva 6 months in artificial saliva at 37 °C five hours in 10% NaOCl and then one hour in distilled water at room temperature	Multiple layer application	Dentin μTBS	iBond SE (Heraeus-Kulzer, Hanau, Germany) Xeno V+ (Dentsply DeTrey, Konstanz, Germany) Scotchbond Universal (3M ESPE) Clearfil S3 Bond (Kuraray Noritaken)
Ugurlu, 2020	24 h of water storage at 37 °C	Multiple layer application	Dentin μTBS	Single Bond Universal (3M ESPE) Gluma Bond Universal (Heraeus Kulzer) Prime&Bond Elect (Dentsply DeTrey) Clearfil SE Bond (Kuraray Noritake)
Zecin-Deren, 2020	24 h of saline storage	Multiple layer application	Dentin SBS	Single Bond Universal (3M ESPE) Prime & Bond One Select (Dentsply) Xeno V (Dentsply) AdperTM Easy One (3M ESPE)
Zecin-Deren, 2020 (b)	24 h of saline storage	Multiple layer application	Dentin SBS	Single Bond Universal (3M ESPE) Prime & Bond One Select (Dentsply) Xeno V (Dentsply) AdperTM Easy One (3M ESPE)
Ayres, 2017	24 h of water storage at 37 °C 2 years of water storage at 37 °C	Non-thermal atmospheric plasma	Dentin μTBS	Scotchbond Universal (3M ESPE)
Ayres, 2018	24 h of water storage at 37 °C 1 years of direct water exposure 1 year of simulated pulpal pressure	Non-thermal atmospheric plasma	Dentin μTBS	Scotchbond Universal (3M ESPE)
Alqahtani, 2014	24 h of water storage at 37 °C	Prolonged curing time	Dentin SBS	Single Bond Universal (3M ESPE) Xeno V+ (Dentsply) AdheSE One F VivaPen (Ivoclar Vivadent)
Sampaio, 2017	24 h of water storage at 37 °C 2 years of water storage at 37 °C	Prolonged curing time	Dentin μTBS	Adper Single Bond 2 (3M ESPE) Optibond Solo Plus (Kerr) Optibond All-In-One (Kerr) Clearfil SE Bond (Kuraray Noritake) Scotchbond Universal (3M ESPE)
Fu, 2017	24 h of water storage at 37 °C	Prolonged blowing time	Dentin μTBS	Scotchbond Universal (3M ESPE) All Bond Universal (Bisco Inc.) G-Premio Bond (GC Corp) Clearfil Universal Bond (Kuraray Noritake) Optibond All-in-one (Kerr)
Luque-Martinez, 2014	24 h of water storage at 37 °C	Prolonged blowing time	Dentin μTBS	Scotchbond Universal (3M ESPE) All Bond Universal (Bisco Inc.) Prime & Bond Elect (Dentsply)
Saikaew, 2018	24 h of water storage at 37 °C 1 year of water storage	Prolonged blowing time	Dentin μTBS	Scotchbond Universal (3M ESPE) G-Premio Bond (GC Corp) Clearfil Universal Bond (Kuraray Noritake)
Irmak, 2018	24 h of water storage at 37 °C 10,000 cycles between 5 °C and 55 °C	Scrubbing technique	Dentin μTBS	Single Bond Universal (3M ESPE) Clearfil Universal Bond Quick (Kuraray Noritake)
Jang, 2018	24 h of water storage at 37 °C Artificial aging (10% aqueous sodium hypochlorite solution for 1 h at room temperature)	Scrubbing technique	Dentin μTBS	G-Premio Bond (GC Corp)
Thanatvarakorn, 2016	24 h of water storage at 37 °C	Scrubbing technique	Dentin μTBS	SE One (Kuraray Noritake) Scotchbond Universal (3M ESPE)
Kharouf, 2019	24 h of water storage at 37 °C	Selective dentin etching	Dentin μTBS	Prime & Bond^®^ active (Dentsply)
Stape, 2018	24 h of water storage at 37 °C 6 months of storage in artificial saliva	Selective dentin etching	Dentin μTBS	Scotchbond Universal (3M ESPE) Scotchbond Multi-Purpose Plus (3M ESPE)
Takamizawa, 2016	24 h of water storage at 37 °C	Selective dentin etching	Dentin SBS	Prime & Bond Elect (Dentsply) Scotchbond Universal (3M ESPE) G-ænial Bond (GC Corp.) OptiBond XTR (Kerr Co.)
Pashaev,2017	24 h of water storage at 37 °C 6 months of water storage at 37 °C	Prolonged application time	Dentin μTBS	All Bond Universal (Bisco Inc.) Single Bond Universal (3M ESPE) Adper Easy One (3M ESPE, St Paul, MN, USA) Adper Single Bond 2 (3M ESPE, St Paul, MN, USA)
Ahmed, 2019	1 week of water storage at 37 °C 6 months of water storage at 37 °C	Prolonged application time	Dentin μTBS	Scotchbond Universal (3M ESPE) Clearfil Universal Bond Quick (Kuraray Noritake) Clearfil SE Bond 2 (Kuraray Noritake)
Amsler, 2015	24 h of water storage at 37 °C	Shortened application time	Dentin SBS	Syntac Classic (Ivoclar Vivadent; Schaan, Liechtenstein) OptiBond FL (Kerry) Clearfil SE Bond (Kuraray Noritake) AdheSE (Ivoclar Vivadent) Xeno Select (Dentsply) Scotchbond Universal (3M ESPE)
Huang, 2017	24 h of water storage at 37 °C 10,000 thermal cycles (10 °C for one min, 25 °C for one min and 55 °C for one min) and 240,000 mechanical cycles	Shortened application time	Dentin μTBS	G-Premio Bond (GC Corp)
Saikaew, 2016	24 h of water storage at 37 °C	Shortened application time	Dentin μTBS	G-Premio Bond (GC Corp) Clearfil Universal Bond (Kuraray Noritake) Scotchbond Universal (3M ESPE)
Saikaew, 2018	24 h of water storage at 37 °C 1 year of water storage at 37 °C	Shortened application time	Dentin μTBS	G-Premio Bond (GC Corp) Clearfil Universal Bond (Kuraray Noritake) Scotchbond Universal (3M ESPE)
Zecin-Deren, 2019	24 h saline solution	Shortened application time	Dentin SBS	Single Bond Universal (3M ESPE) Prime & Bond One Select (Dentsply) Xeno V (Dentsply) AdperTM Easy One (3M ESPE)
Guarda, 2020	24 h of water storage at 37 °C	Electric-current application	Dentin μTBS	Single Bond Universal (3M ESPE) Adper Single Bond 2 (3M ESPE) Clearfil SE Bond (Kuraray Noritake)
Cecchin, 2018	24 h of water storage at 37 °C	a-hydroxy glycolic acid (GA) as a surface pretreatment	Dentin and enamel μTBS	Scotchbond Universal (3M ESPE) Adper Single Bond (3M ESPE) One Step Plus Bisco (Schaumburg)
Cangul, 2020	24 h of water storage at 37 °C	Ozone as cavity disinfectant	Dentin SBS	Clearfil SE Protect primer (Kuraray) Clearfil SE Protect Bond (Kuraray) Peek Universal (Ultradent Products Inc.) Gluma 2 Bond (Heraeus Kulzer)
Demirel, 2019	10,000 thermocycles between 5 °C and 55 °C	Different etching modes and etching time	Dentin μSBS	Single Bond Universal (3M ESPE) All Bond Universal (Bisco Inc.) Clearfil Universal Bond Quick (Kuraray Noritake) Clearfil SE Bond (Kuraray Noritake) Clearfil S3 Bond Plus (Kuraray Noritake) Adper Single Bond 2 (3M ESPE)
Akarsu, 2019	2 h of water storage at 36 °C	Pre-warming of adhesive bottles	Dentin SBS	All Bond Universal (Bisco Inc.) Single Bond Universal (3M ESPE) Clearfil SE Bond (Kuraray Noritake)
Chen, 2020	24 h of water storage at 37 °C 100 days of water storage at 37 °C	Air-blowing with warm air	Dentin μTBS	All Bond Universal (Bisco Inc.) Single Bond Universal (3M ESPE) Clearfil Universal Bond (Kuraray Noritake) Gluma Bond Universal (Heraeus Kulzer) Adhese Universal (Ivoclar Vivadent)
Thanatvarakorn, 2018	24 h of water storage at 37 °C	Smear layer deproteinizing	Dentin μTBS	SE One (Kuraray Noritake) Scotchbond Universal (3M ESPE) BeautiBond Multi (Shofu) Bond Force (Tokuyama Dental)

## Data Availability

The data that support the findings of this study are available from the corresponding author upon reasonable request.

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
