# Peer review of "Bond Strength of Universal Adhesives to Dentin: A Systematic Review and Meta-Analysis"

_polymers, 2021, doi:10.3390/polym13050814_

Round 1

Reviewer 1 Report

Following are some suggestion for the improvement of this review paper; 

  1. Authors are suggested to shorten the title. "of Technique Protocols" can be removed.
  2. Authors are suggested to use updated PRISMA guidelines. please check according to the new guidelines 2020.
  3. Two reviewers independently screened the articles. What was the inter-agreement score according to Cohens kappa? did author perform it?
  4. Authors should make a table of articles excluded after full text evaluation. and reasoning (may be as an appendix) 
  5. Most of the forest plots showed moderate to high heterogeneity. Authors should specify this in the limitations at the end of discussion section.
  6. For forest plots with more than 10 studies, authors should display funnel plot to show explain publication bias.
  7. Manuscript appears to be too lengthy as there is some repetition of the text; please check

Author Response

We were pleased to know that our manuscript was rated as potentially acceptable for publication in the Polymers Journal. We have revised the manuscript guided by referee’s suggestions. Point-by-point responses are below in blue. Modifications performed in the manuscript were highlighted in yellow. In our opinion, the revised manuscript reflects a significant improvement, we wish to thank you for your efforts.

Following are some suggestion for the improvement of this review paper;

  1. Authors are suggested to shorten the title. "of Technique Protocols" can be removed.

R. We appreciate your kind comment. The title was corrected according to your recommendation.

Text change:

Title: Bond strength of universal adhesives to dentin: A systematic review and meta-analysis.

  1. Authors are suggested to use updated PRISMA guidelines. Please check according to the new guidelines 2020.

R. Thank you for the suggestion. We updated the review according to the new PRISMA guidelines adding some new information.

  1. Two reviewers independently screened the articles. What was the inter-agreement score according to Cohens kappa? did author perform it?

R. Yes, the inter-examiner agreement was quantified using the kappa coefficient, and the inter-examiner agreement was excellent (kappa coefficient = 0.87). This information was included in the manuscript. Thank you for the observation!

Text change:

2.1. Literature Search

…The following four electronic databases were screened to identify manuscripts that could be included: PubMed (MedLine), ISI Web of Science, Scopus, and Embase. The inter-examiner agreement was quantified using the kappa coefficient…

  1. Results

Thus, 61 studies were included in the meta-analysis. The inter-examiner agreement was excellent (kappa coefficient = 0.87).

  1. Authors should make a table of articles excluded after full text evaluation. and reasoning (may be as an appendix)

R. Thank you for the comment. A Table with the information requested was added as supplementary material (Appendix A).

  1. Most of the forest plots showed moderate to high heterogeneity. Authors should specify this in the limitations at the end of discussion section.

R. Thank you for the observation. This point was highlighted at the end of the discussion section. 

Text change:

 Discussion.

The methodological quality assessment revealed that most studies included were classified with medium risk of bias, which indicate that the quality of the evidence of the outcome evaluated could be high. In relation to this, it should be highlighted that blinding and sample size calculation were not reported in most of the studies analyzed, and failure to describe these parameters could increase the likelihood of performance, attrition, and detection bias [183].

  1. For forest plots with more than 10 studies, authors should display funnel plot to show explain publication bias.

R. Thanks for the suggestion. Funnel plots for forest plots with more than 10 studies were added.

  1. Manuscript appears to be too lengthy as there is some repetition of the text; please check.

R. Manuscript was shortened according to your suggestion. Thank you!

Reviewer 2 Report

The topic is very interesting and very current. The theme of adhesives is always a topic of great interest in the scientific and clinical community, especially universal adhesives that have so few clinical studies.

Introdution
The article is very well structured. The introduction addresses the topic of generality.

Materials and methods
The materials and methods are well defined. However, there are 2 gaps that authors should change:
1. This review is not registered with PROSPERO, which is recommended.
2. Databases should be more comprehensive. The authors can included the COCHRANE and EMBASE databases.
The inclusion criteria are well-defined, but there is one (the English language) that should be designated as a search filter and not in the inclusion criteria.

Results
Authors should analyze studies with human teeth separately from bovine teeth. In fact, for greater robustness of the review and to avoid doubts about the substrate bias, studies in bovine teeth should be excluded, since the majority are in human teeth.
The authors report little data on the methodology of the studies. Table 2 should include other data from the studies, namely: the type of substrate, the sample size, the type of tooth storage, as they are indicators that can interfere with the performance of adhesives.
Discussion and conclusions
They are adequate and sufficiently extensive, even addressing the limitations of this study. However, the results of the studies' bias analysis are not discussed. The authors should include this analysis in the discussion.

Author Response

We were pleased to know that our manuscript was rated as potentially acceptable for publication in the Polymers Journal. We have revised the manuscript guided by referee’s suggestions. Point-by-point responses are below in blue. Modifications performed in the manuscript were highlighted in yellow. In our opinion, the revised manuscript reflects a significant improvement, we wish to thank you for your efforts.

Materials and methods

The materials and methods are well defined. However, there are 2 gaps that authors should change:

  1. This review is not registered with PROSPERO, which is recommended.

R. We agree with your comment, and we believe that a registration of the systematic review in the PROSPERO database is important, however, PROSPERO only accepts registration of systematic reviews of human studies or systematic reviews of animal studies relevant to human health. Our systematic review is only based on in-vitro studies, therefore, registration in PROSPERO is not possible.

    2. Databases should be more comprehensive. The authors can included the COCHRANE and EMBASE databases.

R. In the revised version of the manuscript, we now included EMBASE within the databases screened. With regards to The Cochrane Library, as we know, this database focuses the most on clinical trials, and considering that this review focused only in in vitro studies, we decided to not include it.

3. The inclusion criteria are well-defined, but there is one (the English language) that should be designated as a search filter and not in the inclusion criteria.

We agree with your comment. The language criterium was used as a search filter. Thank you!

4. Authors should analyze studies with human teeth separately from bovine teeth. In fact, for greater robustness of the review and to avoid doubts about the substrate bias, studies in bovine teeth should be excluded, since the majority are in human teeth.

We accept your suggestion and only studies where human teeth were used as a substrate were included into the review. Thank you!

5. The authors report little data on the methodology of the studies. Table 2 should include other data from the studies, namely: the type of substrate, the sample size, the type of tooth storage, as they are indicators that can interfere with the performance of adhesives.

According to your previous suggestion, we excluded studies where bovine teeth were used as a bonding substrate, therefore, we consider that including the type of substrate within Table 2 is not necessary (as all studies now included evaluated bond strength only on human substrate). Additionally, sample size can be observed directly from the forest plot, so we believe that placing this information in a Table would be repetitive. Information regarding the storage conditions was added. Thank you for the feedback!

Discussion and conclusions

6. They are adequate and sufficiently extensive, even addressing the limitations of this study. However, the results of the studies' bias analysis are not discussed. The authors should include this analysis in the discussion.

Thank you for the observation. This point was highlighted at the end of the discussion section.

Text change:

Discussion.

The methodological quality assessment revealed that most studies included were classified with medium risk of bias, which indicate that the quality of the evidence of the outcome evaluated could be high. In relation to this, it should be highlighted that blinding and sample size calculation were not reported in most of the studies analyzed, and failure to describe these parameters could increase the likelihood of performance, attrition, and detection bias.